# Research on reconfigurable topology layered equalization method based on maximum capacity utilization

**Lingying Tu, Maosheng Xie** *

School of Electrical and Electronic Engineering, Hubei University of Technology, Wuhan, China

* foreverxeiryc@163.com

**Data Availability Statement:** All relevant data are within the paper and its Supporting Information files.

**Funding:** This work was supported by the National Natural Science Foundation of China (41601399).

## Abstract

To maximize the driving range of electric vehicles, battery imbalance is the primary factor that hinders the full utilization of battery pack capacity. This article is based on a four-switch reconfigurable topology, which can flexibly connect, bypass, and parallel any battery cell within the module, and can maintain low voltage fluctuations without the need for a DC-DC converter. Based on this topology, a hierarchical equilibrium strategy combining inter-module K-means clustering analysis and intra-module splitting and recombination is proposed. This strategy can achieve full cell balance, thereby ensuring the maximum capacity utilization of the battery pack. The topology structure composed of 8 batteries was validated, and the experimental results confirmed that the proposed hierarchical balancing strategy supported by the reconfigurable topology increased the capacity utilization of the battery pack by 15.93%, and the maximum fluctuation rate of the battery pack terminal voltage was 0.9%.

## Introduction

In recent years, reconfigurable battery packs have been gained more and more attention due to their good application prospects, and their most prominent advantages are high energy utilization and the ability to dynamically reconfigure the battery cell topology according to the demand of the battery pack. The reconfigurable topology and the control strategy matched with it provide solutions to the problems of low energy transfer efficiency, short lifetime, and low reliability of battery packs [1, 2]. The available battery pack capacity depends on the weakest cell, and the system is cut off while the weakest cell reaches a defined threshold.

Passive or active equalization is usually used to reduce the battery capacity inconsistency. Passive equalization thermally consumes the excess charge of large capacity cells through built-in resistors of shunt, however, this reduces the efficiency of energy utilization [3]. Active equalization uses conversion elements such as transformers, capacitors, and inductors for directional charge transfer [4, 5]. Despite the high efficiency of active equalization, it is large, its cost is high, the speed of equalization is limited, and the layout is complex. Common to both passive and active equalization is the charge transfer between the battery and circuit components. This results in low parasitic power on all electrical components, high energy losses,

The funder had no role in study design, data collection analysis, decision to publish, or preparation of the manuscript.

**Competing interests:** The authors have declared that no competing interests exist.

and accelerated battery degradation due to the presence of additional equalization charge/discharge fluxes [6].

Active equalization, mainly including capacitive-based, inductive-based, transformer-based and DC/DC converter-based, etc. The battery in literature [7] suffers from energy loss during equalization, and the loss increases the more times the charge is transferred. To address this problem, literature [8] proposes a reconfigurable topology that changes the state by switching to make the battery accessible or bypass. By changing the charging and discharging time of the battery cells, the equalization efficiency is improved by reducing the unnecessary charge transfer. Literature [9] presents the problem of adding DC-DC converters within each battery pack thus stabilizing the battery pack but increasing the cost and control difficulty. Literature [10] simplifies the voltage stabilization scheme by adding a redundant unit within each battery pack to be activated alternately in time to achieve this goal.

In this paper, a hierarchical equalization method based on a four-switch topology is proposed to maximize the capacity of the battery pack. The topology has high flexibility, high fault tolerance, and low voltage fluctuation characteristics, while the system complexity can be kept at a reasonable level. The strategy is a hierarchical one that combines inter-module and intra-module equalization, and the equalization of the whole battery cells is achieved by inter-module K-means clustering analysis and intra-module ordered splitting and reorganization, this method not only realizes the voltage stabilization problem of the battery pack but also achieves the maximum capacity utilization of the battery pack.

## Model and topology analysis

To reliably verify the equalization performance, a typical parallel-connected battery module was modeled as shown in Fig 1. Each cell is represented by a well-validated second-order RC equivalent circuit model, and the governing equations of the model are given by [11].

### Equivalent model analysis

$$\frac{dSOC}{dt} = \frac{I}{3600C_n}$$
$$\frac{dU_i}{dt} = -\frac{U_i}{R_iC_i} + \frac{I}{C_i}, i = 1, 2. \tag{1}$$
$$U_t = U_{OC}(SOC) + U_1 + U_2 + R_0I$$

where I, $R_0$, $U_t$, and $C_n$ are the cell current, ohmic resistance, terminal voltage, and nominal capacity, respectively. SOC is the abbreviation for the state of charge, $U_{OC}$ is the open-circuit voltage (OCV), and $R_i$, $C_i$, and $U_i$ ( = 1,2) are the polarization resistance of the first $i$ cell RC pair on both sides, the capacitance, and the voltage.

### Four-switch topology

The number of switches equipped per cell unit determines the diversity of configurations supported by the reconfigurable topology. Various reconfigurable topologies with 1–6 switches per cell have been proposed in the literature [12]. Referring to the existing working requirements, a topology that can freely switch the flexibility of series/parallel connections requires at least three switches. The two most commonly used topologies with three switches (hereafter referred to as topology a and topology b) are shown in Fig 2. Despite their compact space and simple structure, they have very limited configuration flexibility. For example, the independent cell-controlled switches of topology a are only suitable for fully parallel configurations. In

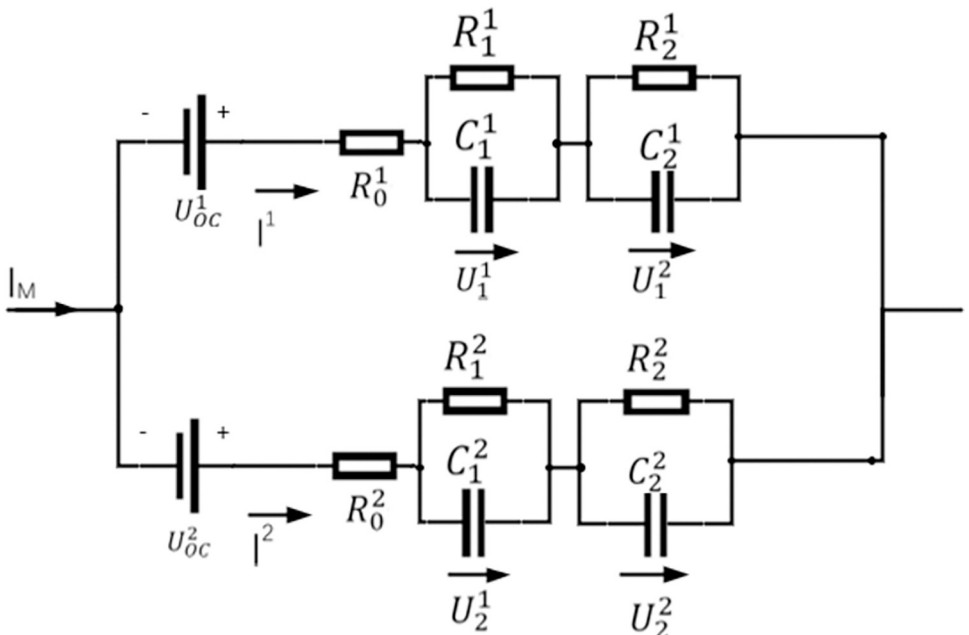

**Fig 1. Parallel model consisting of 2 cells.**

contrast, while topology b extends the applicability to parallel-series configurations, it is not possible to realize independent control of parallel units.

The large number of switching devices makes reconfigurable battery packs subject to a huge short-circuit risk, and the path combination optimization faces the "dimensionality catastrophe" problem, which seriously limits the application of reconfigurable battery packs in large-scale energy storage systems [13]. Typically, topologies with five or more switches can achieve complete flexibility, but further increasing the number of switches beyond three may

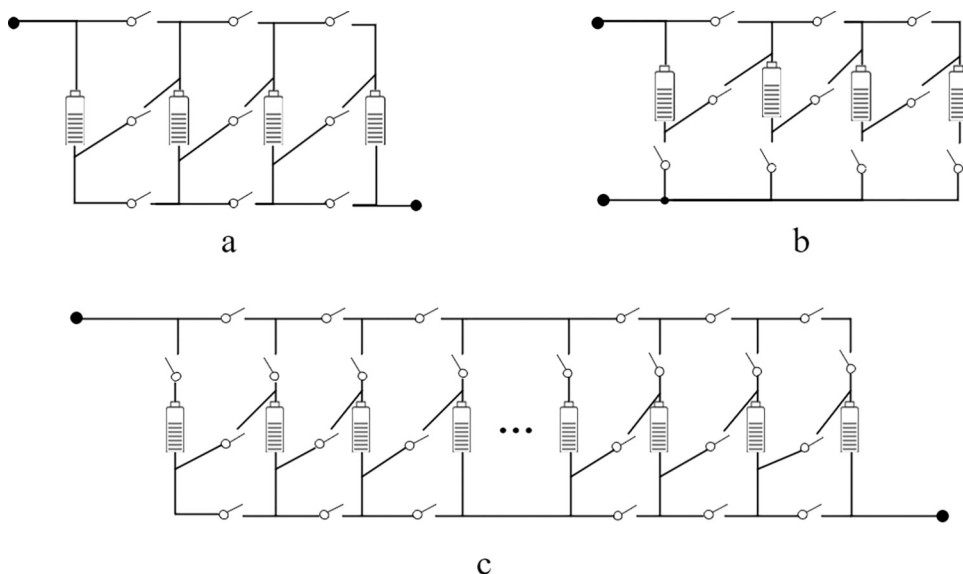

**Fig 2. Topologies a, b three-switch topology, c four-switch topology.**

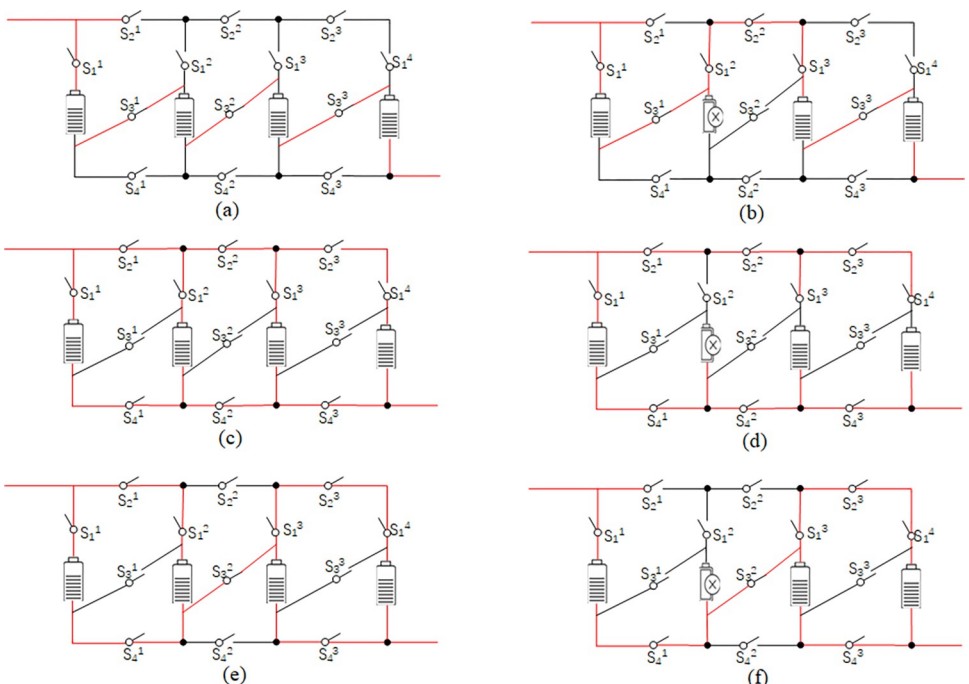

**Fig 3. Working mode of the four-switch topology.**

lead to an exponential increase in complexity. To fulfill the need for sufficient flexibility and affordable control effort, a four-switch reconfigurable topology as shown in Fig 2(C) can fulfill the requirements.

Compared to existing topologies with four or fewer cross switches, this topology can easily isolate faulty battery cells. Through the combination of on and off states of the switches, the proposed topology can be easily realized to cover a wide range of connections such as series, bypass, parallel, and any other combinations. From the practical application point of view, the realized configuration flexibility can meet most of the requirements. As shown in Fig 3(B), switch S2k provides an alternate current path in case of series battery failure. Concerning a single fault in a parallel battery, switch S1k ensures that the faulty battery is disconnected in time before the fault spreads. This is seen in the isolation mechanisms shown in Fig 3(D) and 3(F), and this operation realized by this topology enhances the fault tolerance of the battery pack.

The following is a diagram of Fig 3 Series, Parallel, Mixed, and their fault-tolerant switch connections. (a) 4 cells are connected in series. (b), (a) Fault-tolerant connection. (c) 4 cells are connected in parallel. (d), (c) Fault-tolerant control. (e) Parallel-series connection. (f), (e) Fault-tolerant control.

## Tiered equalization strategy

Battery pack capacity maximization requires all cells to be equalized, i.e., all cells reach an equalized state by the end of operation. It is well known that the parallel-series configuration is a popular choice for automotive applications [14]. Moreover, the equalization of all cells individually in parallel-series configuration is more complex than in series or parallel configuration. Therefore, the focus will be on the equalization of the parallel-series configuration.

Equalization attempts to address two problems: (1) the imbalance between parallel cells within a module; and (2) the "bucket effect" within a series-connected module limited by the

weakest module. Encouraged by the excellent results of modular control [15], a hierarchical equalization scheme supported by a four-switch reconfigurable topology is used to address these problems and achieve full-cell single-cell equalization. This scheme is summarized as inter-module K-means cluster analysis equalization and intra-module split-bypass-reconfiguration.

## Inter-module equalization based on K-means cluster analysis

**Principles of cluster analysis**Cluster analysis is a kind of unsupervised classification. It divides the data set into different classes based on similarity [16], and categorizes the data with small differences into the same class, with significant differences between the different classes. Cluster analysis includes K-means clustering, hierarchical clustering, second-order clustering, and dynamic clustering [17]. In this paper, inter-module equalization utilizes K-means clustering analysis to perform two-dimensional clustering with the voltage and state of charge (SOC) of battery cells. The steps are as follows:

Step 1: Initialize the samples, select $SOC_{max}$ as the first clustering center $Z_1$ select $SOC_{min}$ as the first clustering center $Z_2$.

$$Z_i = \{SOC_{max}, SOC_{min}\} \; i = 1, 2 \tag{2}$$

Step 2: Calculate the distance from all data points to $Z_1$ and $Z_2$ at the current moment. The data points in this paper are two-dimensional data points consisting of SOC as well as the voltage of lithium-ion batteries, while the distribution of voltage and SOC is different according to the actual situation, so the formula for calculating the distance of the sample points to the clustering center adopts the weighted Euclidean distance:

$$D = \sqrt{\alpha_1 (x_1 - y_1)^2 + \alpha_2 (x_2 - y_2)^2} \tag{3}$$

Where $\alpha_1$, and $\alpha_2$ are the weighting coefficients, $(x_1, x_2)$ and $(y_1, y_2)$ are the coordinates of the two points respectively, while the weighting coefficients need to be full

$$\begin{cases} \alpha_1 + \alpha_2 = 1 \\ \dfrac{\alpha_1}{\alpha_2} = \dfrac{\sigma_1}{\sigma_2} \end{cases} \tag{4}$$

where $\sigma_1$ and $\sigma_2$ denote the SOC of the cell as well as the standard deviation of the voltage, respectively.

Step 3: At the end of the classification, update $Z_1$ and $Z_2$. calculate the new cluster centers $Z_1$ and $Z_2$ by taking the arithmetic mean of the SOCs of the same class of cells according to Eqs (5) and (6).

$$Z_i = \frac{\sum_{n=1}^{N} R_{ni} \, SOC_n}{\sum_{n=1}^{N} R_{ni}} \tag{5}$$

$$R_{ni} = \begin{cases} 0 (\alpha_1 < \alpha_2) \\ 1 (\alpha_1 > \alpha_2) \end{cases} \tag{6}$$

Step 4: Compare the square sum of the distances between the new clustering center and the previous clustering center, when the category relationship $R_{ni}$ of each battery cell no longer changes, stop the iteration and output the category to which each battery cell belongs, otherwise skip to step 2 to continue the execution.

## Implementation of inter-module equalization control strategy

①Detect each cell in the battery pack, take the cell SOC size as the judgment basis, if $SOC_{max}$−$SOC_{min}$ <threshold, it indicates that the consistency between the battery cells is better; if $SOC_{max}$−$SOC_{min}$ >threshold, it indicates that the consistency between the battery cells is worse, and it is necessary to turn on the clustering analysis.

②Reorganization pairing using K-means clustering algorithm to classify the battery modules into two categories;

According to the clustering analysis results the relay switch sends the corresponding instructions, a high SOC module within the battery discharge, and every release a certain amount of power, switch once switch combination. The low SOC battery is bypassed and the switch combination is switched once to access the circuit to realize the gradual consistency of the power between battery cells.

④Monitor the average SOC of battery cells in real time, if $SOC_{max}$−$SOC_{min}$<0.1%, then stop equalization; otherwise jump to ②. Equalization control is mainly divided into the equalization of the sum of total power between battery modules and the gradual consistency between battery cells.

The inter-module equalization and intra-module balancing processes are shown in Fig 4. After obtaining the SOC of the battery cells, analyze the clustering results and issue equalization commands to control the on-off of the relay switch, thus controlling the discharge of the high SOC battery cells and the bypass of the low SOC battery cells until the SOC of all the batteries gradually reaches the same level, and during this process, the low SOC battery cells are gradually accessed to the circuit, and ultimately, all the batteries are accessed to the circuit and the equalization ends. The inter-module equalization threshold of the battery is set to α, and the intra-module equalization threshold is set to β.

## Split reorganization equalization in modules

Literature [18]. The gradual increase in the SOC difference between parallel cells during constant-current discharging indicates that the parallel currents do not satisfy the intensity of spontaneous equalization for major charge consumption. Even at small discharge rates, parallel cell current exchange cannot support spontaneous equalization. Spontaneous equalization is significant only when the parallel cell is idle, but it takes a long idle time to reach the equalized state [19]. These results demonstrate the impracticality of spontaneous equalization of parallel cells in electric vehicle applications and seek the need for intra-module parallel cell enhancement in battery equalization.

In existing reconfiguration-based equalization approaches, tandem modules are treated as indivisible entities. In this case, the reconfigurability of components, especially the flexibility of intra-module units, has not been sufficiently used for modular equalization. Inspired by this, an intra-module split-bypass-reorganization equalization method can solve this problem, simply stated as discharging a high-capacity module with a double-rate current compared to the current on the remaining series-connected module. However, both module splitting and reconfiguration lead to changes in the number of series-connected modules, which can lead to unwanted voltage jumps and efficiency discounts. Therefore, a split-bypass-reorganization equalization is proposed to avoid abrupt changes in the total voltage. Table 1 summarizes the proposed intra-module equalization method.

Fig 5 shows only qualitatively the inter-module splitting-reorganization equalization process between two series-connected modules (the dashed box shows the bypassed, unplugged cells of the circuit). Fig 6 shows the topology switch connections for stages 1 and 2.

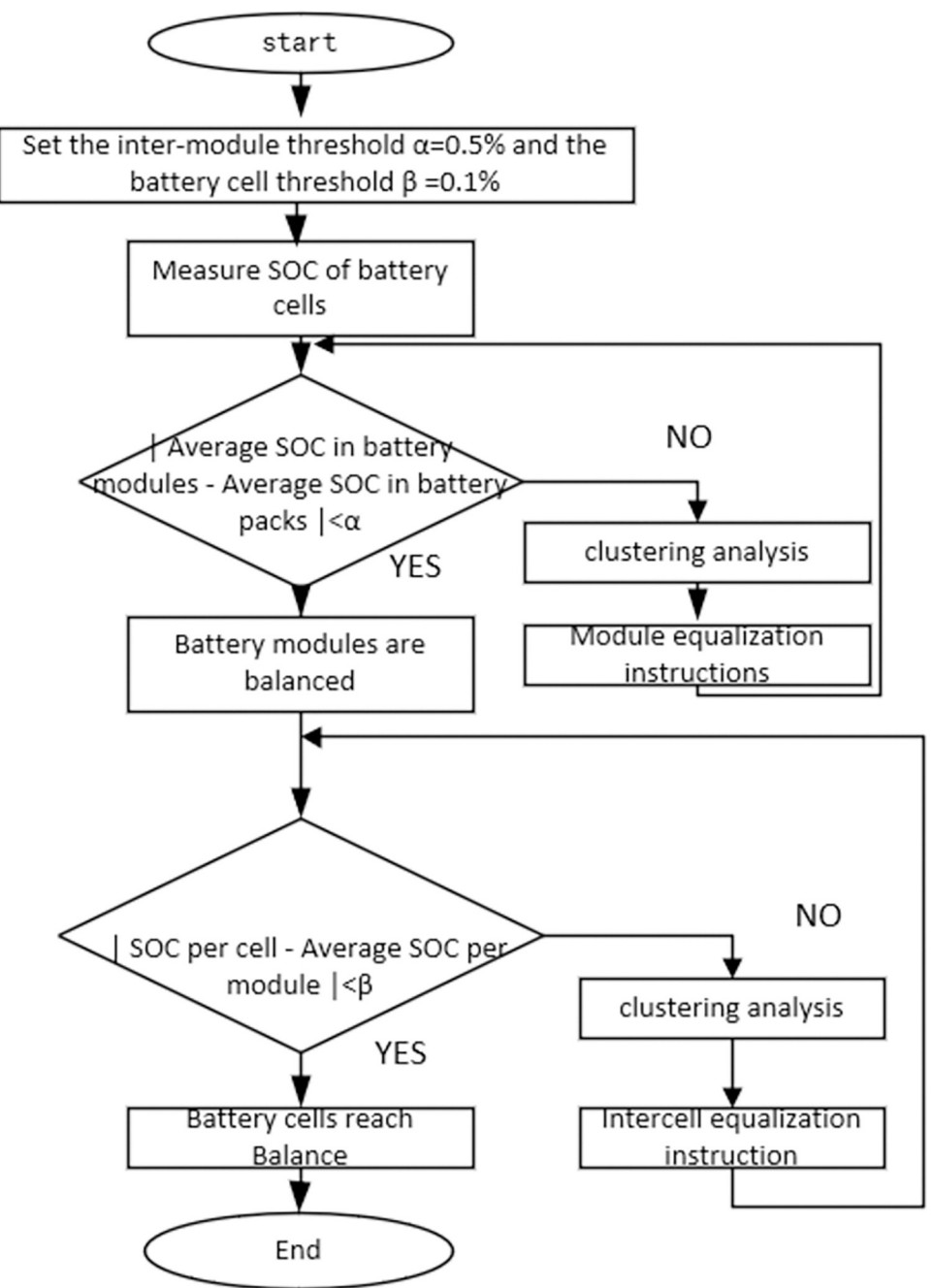

**Fig 4. Time-sharing equalization process.**

The SOC of batteries B1-B8 was discharged to: 48%,49%,51%, 53% 60%, 62%, 64%and 67% before the start of the experiment.

In actual discharges, the rate of capacity reduction of the cells connected in parallel within a module may vary. However, as long as Module B releases 42% more current than Module A (the difference between the total SOC of the four cells within Module B and the total SOC of the four cells within Module A), the goal of series module equalization will be achieved regardless of the capacity values of the cells.

**Table 1. Internal module equalization steps.**

| |
| --- |
| step 1 (parallel to series) |
| 1. The cells of the module with the higher SOC are converted to a series connection to access the circuit so that the original module is split into two series cells. |
| 2. The module or cell with the lower SOC is bypassed to ensure the terminal voltage. |
| Step 2 (Double Discharge) |
| All the modules and sub-modules are discharged in series so that the stronger module or monobloc bears double the discharge current, at which point the weaker module is bypassed and left idle. |
| Step 3 (series to parallel) |
| Once the extra SOC of the stronger module or monobloc is discharged: |
| 1. the module previously converted to series is reconverted to parallel. |
| 2. the bypassed weaker module is connected to the circuit. |

Intra-module balancing can be achieved by using the mechanism of orderly parallel and series conversion of modules. In addition, the exit, idle, and return characteristics of the modules can ensure that the voltage output is stable without large fluctuations during the equalization process.

## Experimental results and analysis

To quantitatively assess the capacity utilization of the proposed equalization scheme, the capacity utilization ratio is defined. The capacity utilization rate is the ratio of the capacity released by all units to the sum of the nominal capacity of all units:

$$P = \sum_{k=1}^{K} C_{end}^{k} \Big/ \sum_{k=1}^{K} C_{n}^{k} \tag{7}$$

where K is the number of cells, and $C_{end}^{k}$ and $C_{n}^{k}$ are the unused capacity at the end of the discharge and the nominal capacity of the Kth (K ∈ [1, K]) cell.

To verify the advantages of the four-switch topology equalization system, the equalization circuit model of eight batteries with this topology is established in the experiment as shown in

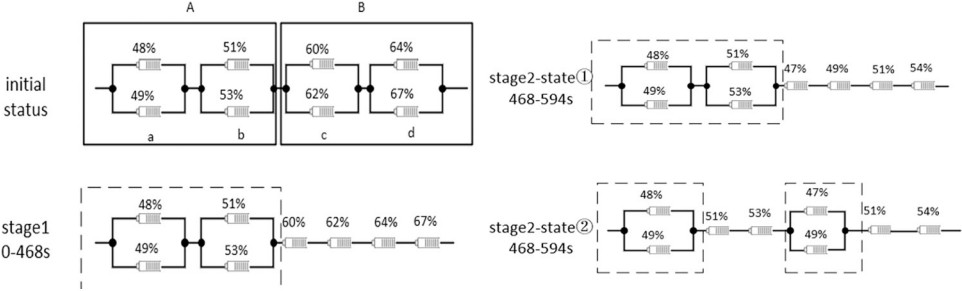

**Fig 5. Inter-module split-reorganization equalization method for four tandem modules.**

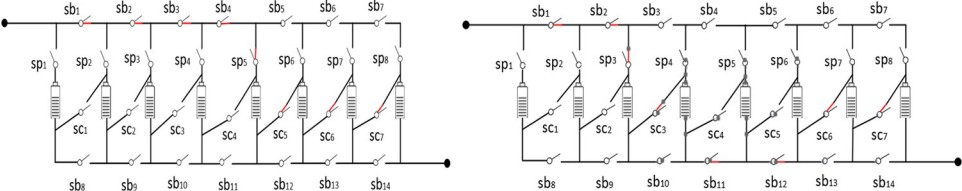

**Fig 6. Topology switch connection diagram for Stage 1 and Stage 2.**

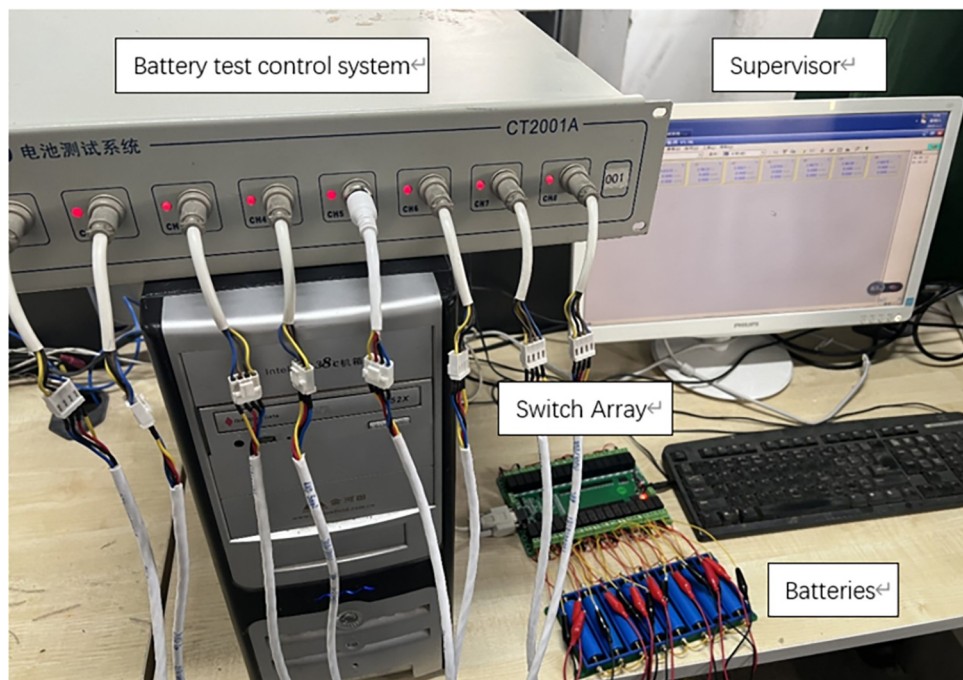

**Fig 7. Battery balancing circuit experiment.**

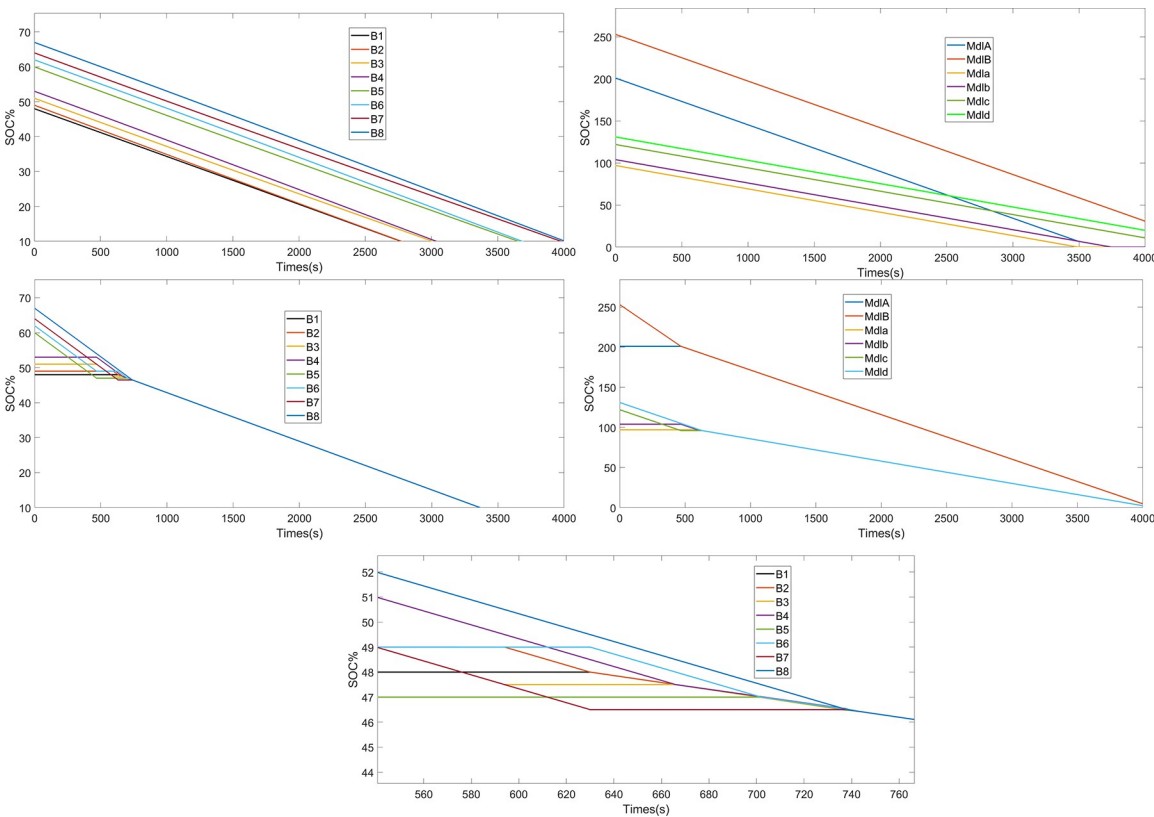

**Fig 8. Experimental results.** (a) No strategy discharge(cells), (b) No strategy discharge(modules), (c) Strategy discharge(cells), (d) Strategy discharge(modules), (e) A localized zoomed-in view of c.

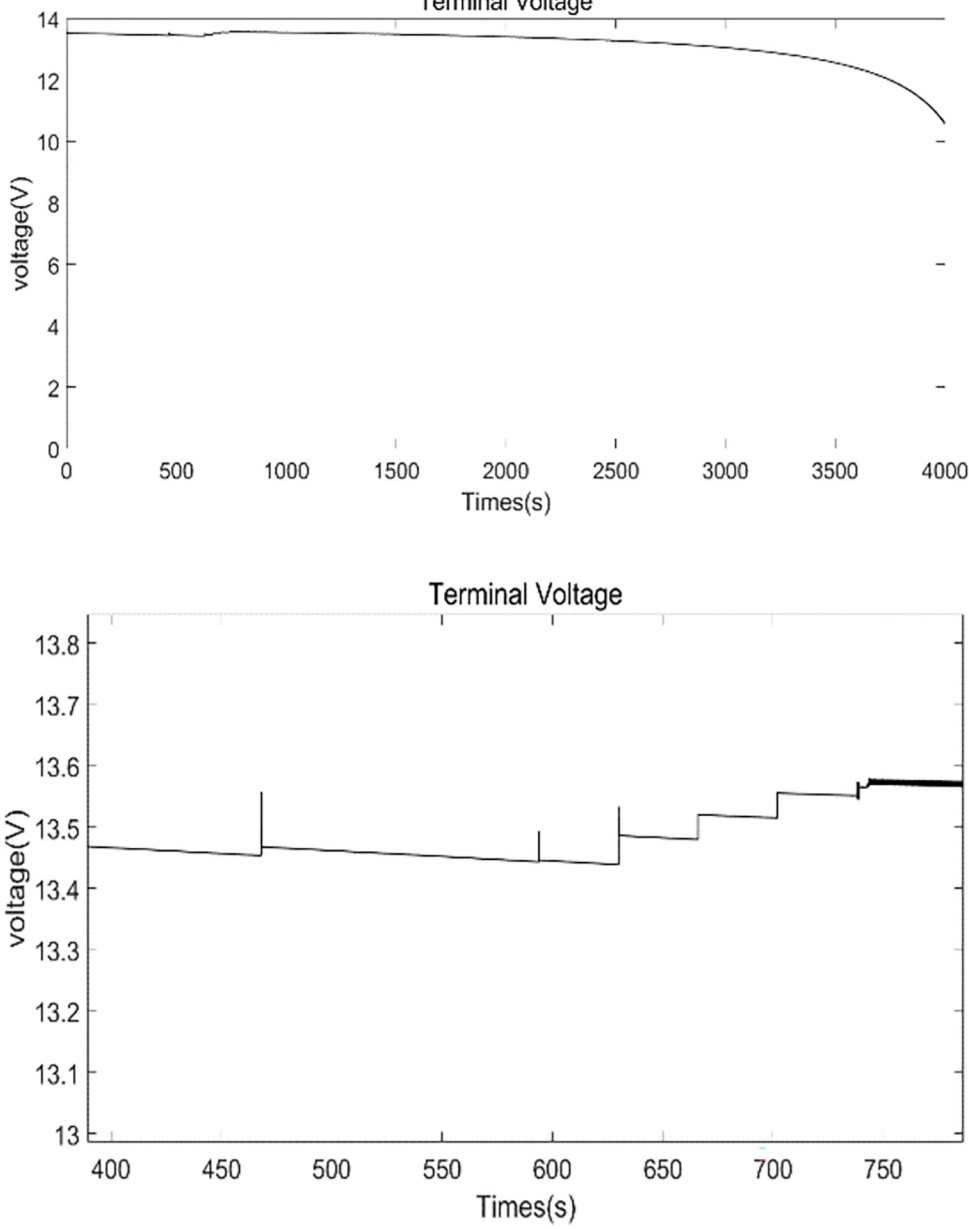

**Fig 9.** Terminal voltage of the battery pack equalization process (a), (b) is the local amplification of (a).

Fig 7, which is equipped with the corresponding control strategy, respectively, and in the reconfigurable equalization circuit, we set the the battery pack discharge at a constant current of 1C.

By comparison, Fig 8(C) and 8(D) show the evolution of the SOC controlled by the proposed hierarchical equalization strategy, and Fig 9 shows the battery pack terminal voltage variation process for the equalization process, with a maximum voltage fluctuation rate of about 0.9% as measured. As shown in Fig 8(D), two phases are observed in the time domain, namely the inter-module equalization phase. In the first inter-module equalization phase from 0–467 s, and the second module equalization phase from 467–597 s, inter-module equalization and battery pack voltage stabilization are achieved simultaneously under the coordination of the

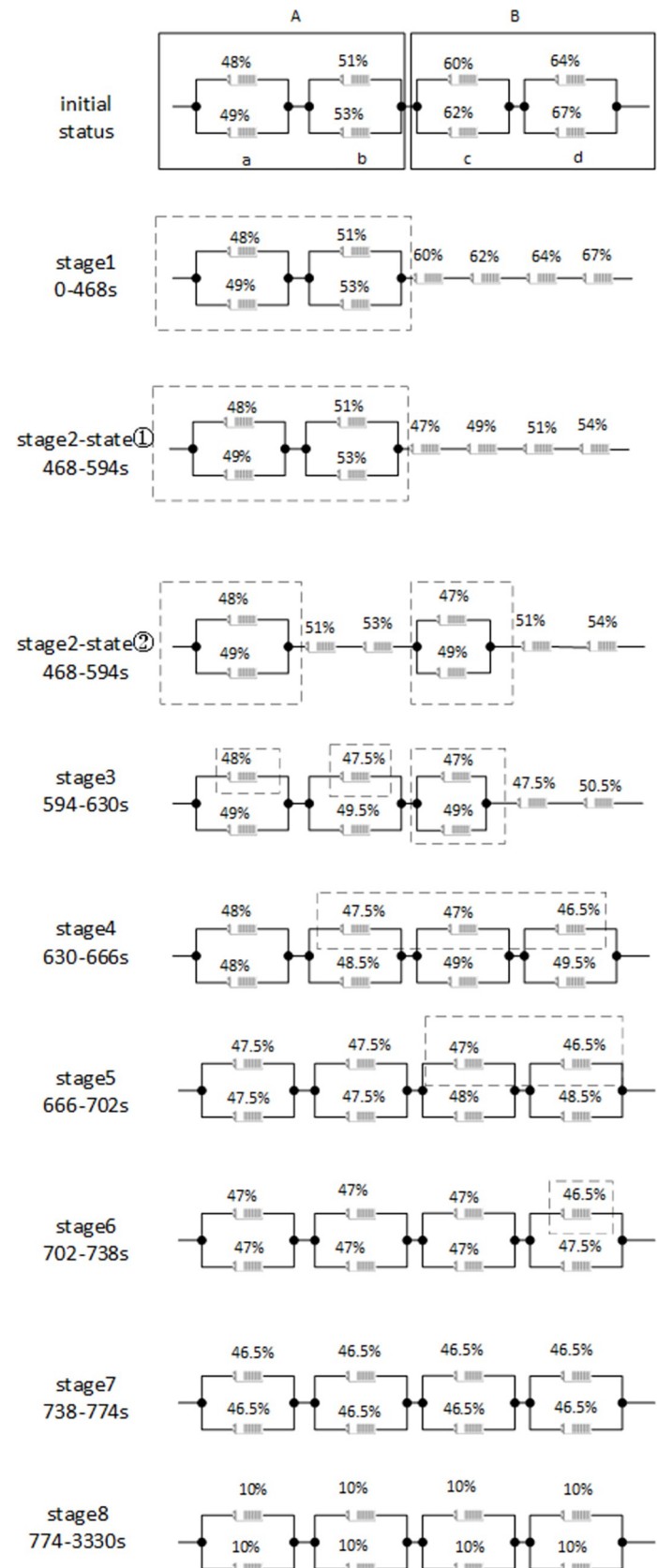

**Fig 10. Configuration of each phase.**

split-bypass-reorganization and exit-re-entry mechanisms. The intra-module equalization phase can be further divided in to four phases, phases 4–7 in Fig 8, in which a weaker cell monomer is split (bypassed) and then recombined, accompanied by the exit and re-entry of a weaker module.

The configuration changes in each stage are shown in Fig 10.

At stage 8 each cell monomer stops discharging and reaches the same discharge termination threshold, whereby all cell monomers release all available power.

$$P_r == \frac{307.890\%}{374} 100\% = 74.07\% \tag{8}$$

where $P_r$ is the capacity utilization without a tiered equalization scheme. In this case study, the lower SOC limit is set to 10%, so the theoretical maximum capacity utilization is 90%. The simulation results reveal that the tiered equalization strategy based on the four-switch reconfigurable topology increases the capacity utilization of the battery pack by 15.93%. Although inter-module equalization is better than intra-module equalization, the capacity utilization rate close to the upper limit indicates that intra-module equalization is also well realized.

## Conclusion

In this paper, a four-switch reconfigurable battery topology and a time-layered equalization control strategy are proposed to control the connection state of every single cell in the battery pack through switches, which not only improves the consistency of the battery pack but also keeps the output voltage of the battery pack relatively stable during the discharge process without the need of a DC/DC converter. The experimental results show that the proposed topology and hierarchical equalization control method are feasible and effective, and significantly improve the utilization efficiency of the battery pack capacity while the operating voltage fluctuation is stabilized within 0.9%.

## Supporting information

**S1 File.**
(XLSX)

**S2 File.**
(DOCX)

## Author Contributions

**Supervision:** Lingying Tu.

**Writing – original draft:** Maosheng Xie.

**Writing – review & editing:** Lingying Tu.

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
