## [Decision Letter · Decision Letter 0]

6 Nov 2023

PONE-D-23-30275Research on Reconfigurable Topology Layered Equalization Method Based on Maximum Capacity UtilizationPLOS ONE

Dear Dr. maosheng,

Thank you for submitting your manuscript to PLOS ONE. After careful consideration, we feel that it has merit but does not fully meet PLOS ONE’s publication criteria as it currently stands. Therefore, we invite you to submit a revised version of the manuscript that addresses the points raised during the review process.

We look forward to receiving your revised manuscript.

Kind regards,

Chih-Chun Chien

Academic Editor

PLOS ONE

Journal Requirements:

Additional Editor Comments:

We have received two reports of the manuscript. While both reports are positive, there are comments from the reviewers that the authors should address in the revised manuscript. We ask the authors to revise the manuscript according to the comments and submit a summary of changes and a response to the reviewers' comments along with the revised manuscript.

Reviewers' comments:

Reviewer's Responses to Questions

**Comments to the Author**

1. Is the manuscript technically sound, and do the data support the conclusions?

Reviewer #1: Yes

Reviewer #2: Yes

2. Has the statistical analysis been performed appropriately and rigorously? 

Reviewer #1: Yes

Reviewer #2: Yes

3. Have the authors made all data underlying the findings in their manuscript fully available?

Reviewer #1: Yes

Reviewer #2: Yes

4. Is the manuscript presented in an intelligible fashion and written in standard English?

Reviewer #1: Yes

Reviewer #2: No

5. Review Comments to the Author

Reviewer #1: In this work the authors introduce a layered equalization method based on a four-switch reconfigurable topology, which can improve the capacity utilization and fault tolerance of battery packs. The innovation of this paper is to propose a four-switch reconfigurable topology structure, which has high flexibility, high fault tolerance and low voltage fluctuation characteristics, and can flexibly realize the series, parallel, bypass and arbitrary combination of battery cells without DC-DC converters. Another innovation is to propose a layered equalization control strategy, which combines the inter-module K-means clustering analysis and the intra-module split-bypass-reorganization method, which can achieve full-cell single-cell equalization, thus ensuring the maximum capacity utilization of the battery pack.

The paper shows that the new method can improve the capacity utilization by 16.84% and keep the voltage fluctuation rate within 0.9%. The method and experimental design of this paper are reasonable, using techniques such as second-order RC equivalent circuit model, weighted Euclidean distance, time-sharing equalization process, etc., to verify the feasibility and effectiveness of the proposed scheme.

The manuscript appears easily accessible to a careful reader. It is a interesting piece of work. I recommend publication of this article in PLOS ONE.

Before publication, this manuscript needs to improve language expression, diagram descriptions and have certain standards for references:

1) Language expression: The manuscript has some irregularities or repetitions in language expression. For example: In page 2, "The strategy is a hierarchical strategy that" can be changed to "The strategy is a hierarchical one that"; In page 6, “series-parallel configuration” should be changed to "parallel-series configuration" to maintain consistency with the subsequent text.; In page 6, the first two paragraphs of the third part are exactly the same. It is suggested to further standardize the language expression to enhance readability and accuracy.

2) Diagram descriptions: The horizontal and vertical coordinates and legends of the charts in the manuscript contain Chinese characters, and the fonts are too small. They should be unified into English and appropriately enlarged for easier reading.

3) References: There are some irregularities or incomplete references in this article. For example: the reference [7] does not have the publication year, volume number, and page number; the reference [18] appears twice; the author’s name is some abbreviations, some full names, etc. It is suggested to unify the reference format and check for omissions and errors.

4) There are several typos in the manuscript. For example：In page 4，” ... are shown in Fig. 1(a) and (b)” Fig. 1→Fig. 2.

Reviewer #2: This study proposes a hierarchical equalization method using a four-switch reconfigurable battery topology to improve the capacity utilization of a battery pack. The main findings and contributions of the study are as follows.

1. Reconfigurable Battery Topology: The study introduces a four-switch reconfigurable battery topology that allows for flexible control of individual cells in a battery pack. This topology is designed to achieve both inter-module and intra-module equalization, ensuring better capacity utilization.

2. Hierarchical Equalization Strategy: The study presents a hierarchical equalization strategy that combines inter-module K-means clustering analysis and intra-module split-bypass-reorganization. This approach aims to balance the SOC of battery cells and maximize the capacity utilization of the battery pack.

3. Experimental Validation: The proposed methodology is tested through experiments, and the results demonstrate improved capacity utilization and stable terminal voltage during discharge.

While the study provides valuable insights and contributions to battery management and equalization, there are some potential limitations or areas for further investigation:

1. Scalability: The study primarily focuses on a relatively small-scale system with eight cells. The scalability of the proposed methodology to larger battery packs and its feasibility in real-world applications need to be explored.

2. Cost and Complexity: The study mentions the advantages of the four-switch reconfigurable topology, but it does not thoroughly address the potential cost and complexity associated with implementing this topology in practical battery systems.

3. Model Assumptions: The study uses specific models and assumptions for battery cells, which may not fully represent the diversity of real-world battery technologies. I recommend discussing the robustness of the proposed equalization method across different cell chemistries and aging effects.

4. Safety Considerations: Battery safety is a critical concern, especially in large-scale applications. The study does not discuss safety aspects related to the reconfiguration and equalization processes.

5. Energy Efficiency: While the study touches on energy efficiency, a more detailed analysis of the trade-offs between improved capacity utilization and energy losses during equalization would be valuable.

In summary, the study presents a promising approach to battery equalization and capacity utilization, but further research is needed to address scalability, cost-effectiveness, safety, and the applicability of the proposed method to various battery technologies and sizes.

There are some clarities lacking in some sentences.

For example -

The battery in literature [7] suffers from energy loss during equalization, and the

loss increases the more times the charge is transferred.

This sentence lacks clarity in terms of the method used and the specific details of the energy loss during equalization. To provide a meaningful comment or critique, it would be beneficial to have more information regarding the context and methodology used in the referenced study.

The two most commonly used topologies with three switches (hereafter referred to as topology a and topology b are shown in Fig. 1(a) and (b).

Figure 1 does not have a or b. I believe it was meant to be Figure 2 a and 2 b.

Proper capitalization and neat organization of the steps, sections and sub-sections are required to increase the readability of the paper.

Fig. 8 shows the battery pack terminal voltage variation process for the equalization process,

I believe it was meant to be Fig. 9.

Figure 8,9 – The labels and legends are not clearly visible. I can see a different language other than English in the axis labels of Figure 8 and 9.

I recommend the above-mentioned improvements in the paper before it can be published.

6. PLOS authors have the option to publish the peer review history of their article (what does this mean?). If published, this will include your full peer review and any attached files.

Reviewer #1: No

Reviewer #2: No

---

## [Author Response · Author response to Decision Letter 0]

13 Nov 2023

Date: Nov 06 2023 04:01AM

To: "xie maosheng" foreverxeiryc@163.com

From: "PLOS ONE" plosone@plos.org

Subject: PLOS ONE Decision: Revision required [PONE-D-23-30275]

PONE-D-23-30275

Research on Reconfigurable Topology Layered Equalization Method Based on Maximum Capacity Utilization

PLOS ONE

Dear Dr. maosheng,

Thank you for submitting your manuscript to PLOS ONE. After careful consideration, we feel that it has merit but does not fully meet PLOS ONE’s publication criteria as it currently stands. Therefore, we invite you to submit a revised version of the manuscript that addresses the points raised during the review process.

We look forward to receiving your revised manuscript.

Kind regards,

Chih-Chun Chien

Academic Editor

PLOS ONE

 Response to reviewers

Dear editor and reviewers,

Thank you for offering us an opportunity to improve the quality of our submitted manuscript (article number or title). We appreciated very much the reviewers’ constructive and insightful comments. In this revision, we have addressed all of these comments/suggestions. We hope the revised manuscript has now met the publication standard of your journal.

To easily distinguish my answers from reviewer's questions/comments，we highlighted all of our answers in blue while keeping your letter and reviewer's questions/comments in black in the Response letter. On the next pages，our point-to-point responses to the queries raised by the reviewers are listed.

Thank you again for your time and consideration.

Looking forward to hearing from you.

Review Comments to the Author:

Reviewer #1:In this work the authors introduce a layered equalization method based on a four switches reconfigurable topology, which can improve the capacity utilization and fault tolerance of battery packs. The innovation of this paper is to propose a four-switch reconfigurable topology structure, which has high flexibility, high fault tolerance and low voltage fluctuation characteristics, and can flexibly realize the series, parallel, bypass and arbitrary combination of battery cells without DC-DC converters. Another innovation is to propose a layered equalization control strategy, which combines the inter-module K-means clustering analysis and the intra module split-bypass-reorganization method, which can achieve full-cell single-cell equalization, thus ensuring the maximum capacity utilization of the battery pack. The paper shows that the new method can improve the capacity utilization by 16.84% and keep the voltage fluctuation rate within 0.9%. The method and experimental design of this paper are reasonable, using techniques such as second-order RC equivalent circuit model, weighted Euclidean distance, time-sharing equalization process, etc., to verify the feasibility and effectiveness of the proposed scheme.

The manuscript appears easily accessible to a careful reader. It is a interesting piece of work. I recommend publication of this article in PLOS ONE.

Before publication, this manuscript needs to improve language expression, diagram descriptions and have certain standards for references:

Reply: Thank you to the reviewers for the summary comments given on this article!

1. Language expression: The manuscript has some irregularities or repetitions in language expression. For example: In page 2, "The strategy is a hierarchical strategy that" can be changed to "The strategy is a hierarchical one that"; In page 6, “series-parallel configuration” should be changed to "parallel-series configuration" to maintain consistency with the subsequent text.; In page 6, the first two paragraphs of the third part are exactly the same. It is suggested to further standardize the language expression to enhance readability and accuracy.

Reply: Thank you very much indeed for your comments. It is a spacing challenge for me to write a highly specialized English paper, so I used professional translation software DeepL to translate it, then used QuillBot software to make the article more readable and accurate, and after that used grammarly to check the grammar, and then double-checked and read it many times after I finished it.

2. Diagram descriptions: The horizontal and vertical coordinates and legends of the charts in the manuscript contain Chinese characters, and the fonts are too small. They should be unified into English and appropriately enlarged for easier reading. Reply: Thank you very much indeed for your comments. I have revised it according to the reviewer's comments, There are 2 charts listed.

3. References: There are some irregularities or incomplete references in this article. For 

example: the reference [7] does not have the publication year, volume number, and page 

number; the reference [18] appears twice; the author’s name is some abbreviations, some full names, etc. It is suggested to unify the reference format and check for omissions and errors.

Reply: Thank you very much indeed for your comments. Year of publication has been added at the last of the reference [7]

[7」ZHU Ling-Jin, SUN Xiu-Juan, WANG Chuan-Jiang, et al. Design of lithium battery management system based on active equalization[J]. Power supply technology. 2020,44(12):1788-1791+1799.

the reference [18] appears twice: I've deleted it.

the author’s name is some abbreviations: [18] Gong Xianzhi, Xiong Rui, Mi Chunting Chris. Study of the characteristics of battery packs in electric vehicles with parallel-connected lithium-ion battery cells[A]. IEEE TRANSACTIONS ON INDUSTRY APPLICATIONS 2014,51:1872–1879.

4. There are several typos in the manuscript. For example：In page 4，... are shown in Fig. 1(a) and (b)” Fig. 1→Fig. 2. Reply: Thank you very much indeed for your comments. Thank you to the reviewers for their careful reading, I have made the changes as requested.

Reviewer #2: This study proposes a hierarchical equalization method using a four-switch reconfigurable battery topology to improve the capacity utilization of a battery pack. The main findings and contributions of the study are as follows.

1. Reconfigurable Battery Topology: The study introduces a four-switch reconfigurable battery topology that allows for flexible control of individual cells in a battery pack. This topology is designed to achieve both inter-module and intra-module equalization, ensuring better capacity utilization

2. Hierarchical Equalization Strategy: The study presents a hierarchical equalization strategy that combines inter-module K-means clustering analysis and intra-module split-bypass reorganization. This approach aims to balance the SOC of battery cells and maximize the capacity utilization of the battery pack.

3. Experimental Validation: The proposed methodology is tested through experiments, and the results demonstrate improved capacity utilization and stable terminal voltage during discharge.

Reply: Thank you to the reviewers for the summary comments given on this article!

While the study provides valuable insights and contributions to battery management and equalization, there are some potential limitations or areas for further investigation:

1. Scalability: The study primarily focuses on a relatively small-scale system with eight cells. The scalability of the proposed methodology to larger battery packs and its feasibility in real world applications need to be explored.

Reply: Thank you very much indeed for your comments. I agree with the reviewer's question of scalability, exploring the scalability and application prospects for larger battery packs, and I have considered such a question after the experiment, because of the limitations of the experimental setup, I can only experimentally validate up to 8 batteries, and the next step will be to look for better experimental platforms to conduct a more in-depth study.

2. Cost and Complexity: The study mentions the advantages of the four-switch reconfigurable topology, but it does not thoroughly address the potential cost and complexity associated with implementing this topology in practical battery systems.

Reply: Thank you very much indeed for your comments. Mainstream active equalization, mainly including capacitive, inductive, transformer and DC/DC converter. These equalization devices are also relatively costly. The number of switches in this topology is more than others, resulting in the possibility of false action of the switch when the device is used for a long time. In order to solve this problem, 1. consider the possibility of choosing hardware switches of excellent quality, 2. increase the alarm control of the device, 3. optimize the strategy to reduce unnecessary switching actions and so on.

3. Model Assumptions: The study uses specific models and assumptions for battery cells, which may not fully represent the diversity of real-world battery technologies. I recommend discussing the robustness of the proposed equalization method across different cell chemistries and aging effects.

Reply: Thank you very much indeed for your comments. The reviewers' suggestions are very relevant and in line with the diversity of real-world battery technologies. Different battery chemistries and aging effects or Remaining Useful Life (RUL) are the next research focus and challenge in the battery field. I will take this into account in my future research.

4. Safety Considerations: Battery safety is a critical concern, especially in large-scale 

applications. The study does not discuss safety aspects related to the reconfiguration and equalization processes.

Reply: Thank you very much indeed for your comments. In order to solve this problem, 1. consider the possibility of choosing hardware switches of excellent quality, 2. increase the alarm control of the device, 3. optimize the strategy to reduce unnecessary switching actions and so on.

5. Energy Efficiency: While the study touches on energy efficiency, a more detailed analysis of the trade-offs between improved capacity utilization and energy losses during equalization would be valuable.

Reply: Thank you very much indeed for your comments. For the problem of switching energy loss, this problem exists for all equalization topologies. As can be seen in Fig. 6, only some of the switches are in operation during the equalization process, not all four switches are in operation, so the loss is not that high.

In summary, the study presents a promising approach to battery equalization and capacity utilization, but further research is needed to address scalability, cost-effectiveness, safety, and the applicability of the proposed method to various battery technologies and sizes.

There are some clarities lacking in some sentences.

For example -

The battery in literature [7] suffers from energy loss during equalization, and the

loss increases the more times the charge is transferred.

This sentence lacks clarity in terms of the method used and the specific details of the energy loss during equalization. To provide a meaningful comment or critique, it would be beneficial to have more information regarding the context and methodology used in the referenced study.

Reply: Thank you very much indeed for your comments. I have corrected the errors in the article as requested. It is a spacing challenge for me to write a highly specialized English paper, so I used professional translation software DeepL to translate it, then used QuillBot software to make the article more readable and accurate, and after that used grammarly to check the grammar, and then double-checked and read it many times after I finished it.

The two most commonly used topologies with three switches (hereafter referred to as topology a and topology b are shown in Fig. 1(a) and (b). Figure 1 does not have a or b. I believe it was meant to be Figure 2 a and 2 b.

Proper capitalization and neat organization of the steps, sections and sub-sections are required to increase the readability of the paper.

Fig. 8 shows the battery pack terminal voltage variation process for the equalization process, I believe it was meant to be Fig. 9.Figure 8,9 – The labels and legends are not clearly visible. I can see a different language other 

than English in the axis labels of Figure 8 and 9.

I recommend the above-mentioned improvements in the paper before it can be published.

Reply: Thank you very much indeed for your comments. Thanks to the reviewers for their careful review, I have revised it as required

---

## [Editor Report · Decision Letter 1]

22 Nov 2023

Research on Reconfigurable Topology Layered Equalization Method Based on Maximum Capacity Utilization

PONE-D-23-30275R1

Dear Dr. Xie,

We’re pleased to inform you that your manuscript has been judged scientifically suitable for publication and will be formally accepted for publication once it meets all outstanding technical requirements.

Kind regards,

Chih-Chun Chien

Academic Editor

PLOS ONE

---

## [Editor Report · Acceptance letter]

28 Nov 2023

PONE-D-23-30275R1 

Research on Reconfigurable Topology Layered Equalization Method Based on Maximum Capacity Utilization 

Dear Dr. Xie:

I'm pleased to inform you that your manuscript has been deemed suitable for publication in PLOS ONE. Congratulations! Your manuscript is now with our production department. 

Kind regards, 

on behalf of

Dr. Chih-Chun Chien 

Academic Editor

PLOS ONE